# Method for Suppressing Non-Stationary Interference in the Main-Lobe Based on a Multi-Polarized Array

**DOI:** 10.3390/s25216587

**Published:** 2025-10-26

**Authors:** Jie Wang, Shujuan Ding, Na Wei, Jinzhi Bi, Rongqiu Zheng

**Affiliations:** China Research Institute of Radiowave Propagation, Qingdao 266107, China; 9215wangjie@163.com (J.W.); 13280800008@163.com (N.W.); bjinzhi@163.com (J.B.); zhengrq123@126.com (R.Z.)

**Keywords:** main-lobe interference, multi-polarized array, non-stationary interference, matched filtering, time–frequency analysis, beamforming

## Abstract

To suppress non-stationary main-lobe interference, we utilized the waveform information of the transmitted signal and proposed an interference suppression method based on a multi-polarized array without the need for calculating the target parameters. This method calculates the steering vector of the target through matched filtering. Additionally, for non-stationary interference whose statistical characteristics change over time, we extract high-energy frequency points from the time–frequency joint domain to obtain the time–frequency covariance matrix for subsequent beamforming. Simulation experiments demonstrate that this method leverages the signal polarization information sensed by the multi-polarized array, effectively suppressing non-stationary main-lobe interference in the polarization domain. This method does not require estimation of the target’s polarization parameters and is more suitable for real-world detection scenarios where the waveform is known.

## 1. Introduction

The ionosphere is a highly ionized region in Earth’s atmosphere and serves as an important component of the geospace environment. Temporal and spatial variations in the ionosphere have significant impacts on the propagation of electromagnetic waves in space. Accurate monitoring, prediction, and early warning of the morphology, structure, and disturbances of the ionosphere are crucial in various fields such as short-wave communication, aerospace, navigation, meteorological monitoring, disaster warning, and national defense [1,2,3,4,5].

In modern ionospheric detection, the echo signal is easily affected by the interference in the environment, which has the characteristics of time-varying, non-stationary, multipath propagation, and highly overlaps with the target signal in the time–frequency domain [6,7,8].

Ionospheric interference suppression is the core of ionospheric detection. Current suppression techniques primarily focus on signal propagation mechanisms and multi-dimensional signal processing, such as space-time adaptive processing [9,10,11,12] and time–frequency domain joint processing [13,14,15]. Most existing methods assume that the interference characteristics are relatively stable, and cannot be applied to the ionospheric interference in the actual environment, which presents non-stationary characteristics. In addition, the spectral overlap of short-wave communications, radar signals, etc., with ionospheric interference is still a problem, so it is necessary to study non-stationary interference suppression methods that utilize the characteristics of other domains in addition to the time–frequency domain.

The polarization of electromagnetic waves is another important characteristic that can be utilized, along with their amplitude, phase, frequency, and waveform. Fully utilizing the polarization information of electromagnetic waves can significantly improve the performance of signal processing systems [16,17,18]. A conventional single-polarized array cannot obtain the vector information of electromagnetic wave polarization. When the polarization of the incident signal is orthogonal to the polarization of the single-polarized array, the array will not produce an effective output, causing serious impacts on the subsequent signal processing [19,20]. Utilizing a multi-polarization array for reception can obtain the polarization information of signals. A multi-polarization array consists of sensors with incompletely identical polarization selection characteristics and can observe electromagnetic waves in a vectorial manner. The polarization information obtained in this way provides new degrees of freedom for interference suppression and offers new solutions for handling interferences that cannot be suppressed by conventional methods [21,22,23,24].

Over the past few decades, numerous studies on robust adaptive beamforming have been conducted, such as sample matrix inversion (SMI) beamformer [25], worst-case performance optimization (WCPO) beamformer [26], and the minimum variance distortionless response (MVDR) beamformer [27]. These studies are all based on a single-polarized array and can be easily extended to a multi-polarized array [28]. Furthermore, based on a multi-polarized array, some scholars have also conducted research on correlated sources beamforming [29], wideband signal beamforming [30], sparse array beamforming [31], and so on. However, these methods require the known angle and polarization parameters of the target. Main-lobe interference poses one of the most significant challenges in fields such as radar, communications, and hydroacoustic detection [32,33,34,35,36]. In the presence of main-lobe interference, we cannot easily distinguish the target signal from the interference signal to obtain the parameters of the target. Traditional sidelobe interference suppression methods are difficult to effectively deal with main-lobe interference due to its overlap with target signals in the spatial domain. Furthermore, the existing methods deal with stationary signals whose statistical characteristics do not change over time. There is little research on beamforming methods for non-stationary signals. Motivated by this, this paper proposes a main-lobe interference suppression method based on multi-polarized array reception for ionospheric non-stationary interference without estimation of the target’s parameters. This method utilizes the waveform information of the transmitted signal without requiring estimation of the polarization parameters of the target. It calculates the steering vector of the target during beamforming through matched filtering and obtains a time–frequency covariance matrix through time–frequency analysis and selection of time–frequency points for subsequent beamforming. Simulation experiments have verified the effectiveness of this method.

## 2. Received Signal Model

The signal propagation schematic in a rectangular coordinate system is shown in Figure 1.

Define $b_{S}\left(\theta ,\phi \right)$ as the unit vector pointing in the direction of the incoming wave from the far field, with its opposite direction representing the propagation direction of the signal wave. Let $b_{P}\left(\theta ,\phi \right)$ denote the propagation vector. Then,(1)$$b_{S}\left(\theta ,\phi \right)=-b_{P}\left(\theta ,\phi \right)={\left[\sin\theta \cos\phi ,\sin\theta \sin\phi ,\cos\theta \right]}^{T}$$

In Equation (1),  θ and ϕ represent the pitch angle and azimuth angle of the incoming wave direction, respectively, $0^{{}^{\circ}}\leq \theta \leq 180^{{}^{\circ}}$, $0^{{}^{\circ}}\leq \phi \leq 360^{{}^{\circ}}$; T denotes transposition. $b_{H}\left(\phi \right)$ and $b_{V}\left(\theta ,\phi \right)$ are a pair of standard orthogonal vectors within the polarization plane, referred to as the horizontal polarization basis vector and vertical polarization basis vector, respectively. Where(2)$$b_{H}\left(\phi \right)={\left[-\sin\phi ,\cos\phi ,0\right]}^{T}$$(3)$$b_{V}\left(\theta ,\phi \right)={\left[\cos\theta \cos\phi ,\cos\theta \sin\phi ,-\sin\theta \right]}^{T}$$

Assuming that the elements of a multi-polarized array consist of L electromagnetic vector sensors, which receive a total of M signals with a carrier frequency of $\omega_{0}$, the signal received at the array can be expressed as:(4)$$\begin{array}{ll} \;\;\;\;x\left(t\right) & =\;\sum_{m=1}^{M}\underset{H_{\theta_{m,}\phi_{m}}}{\underbrace{\left[a_{H}\left(\theta_{m},\phi_{m}\right),a_{V}\left(\theta_{m},\phi_{m}\right)\right]}}\underset{s_{H+V,m}\left(t\right)}{\underbrace{\left[\begin{array}{c} s_{H,m}\left(t\right)\\ s_{V,m}\left(t\right) \end{array}\right]}}\;+n\left(t\right)\\ & =H_{\theta_{1},\phi_{1}}s_{H+V,1}\left(t\right)\;+\;\underset{i\left(t\right)}{\underbrace{\sum_{m=2}^{M}H_{\theta_{m},\phi_{m}}s_{H+V,m}\left(t\right)}}\;+\;n\left(t\right) \end{array}$$

In Equation (4), $s_{H,m}\left(t\right)$ and $s_{V,m}\left(t\right)$ represent the horizontally polarized component and vertically polarized component of the mth incident signal, respectively. $n\left(t\right)$ denotes additive white Gaussian noise, $i\left(t\right)$ represents the interference signal. $a_{H}\left(\theta_{m},\phi_{m}\right)$ and $a_{V}\left(\theta_{m},\phi_{m}\right)$ are referred to as the horizontally polarized manifold vector and vertically polarized manifold vector of the mth incident signal, respectively. Assuming that $d_{l}$ is the position vector of the lth array antenna element, and c is the speed of electromagnetic waves in vacuum, the specific forms of $a_{H}\left(\theta_{m},\phi_{m}\right)$ and $a_{V}\left(\theta_{m},\phi_{m}\right)$ are shown below.(5)$$a_{H}\left(\theta_{m},\phi_{m}\right)=\;a\left(\theta_{m},\phi_{m}\right)\;⨂b_{iso-H}\left(\theta_{m},\phi_{m}\right)$$(6)$$a_{V}\left(\theta_{m},\phi_{m}\right)=a\left(\theta_{m},\phi_{m}\right)\;⨂\;b_{iso-V}\left(\theta_{m},\phi_{m}\right)$$(7)$$a\left(\theta_{m},\phi_{m}\right)={\left[e^{j\omega_{0}\tau_{1}\left(\theta_{m},\phi_{m}\right)},e^{j\omega_{0}\tau_{2}\left(\theta_{m},\phi_{m}\right)},\ldots ,e^{j\omega_{0}\tau_{L}\left(\theta_{m},\phi_{m}\right)}\right]}^{T}$$(8)$$\tau_{l}\left(\theta_{m},\phi_{m}\right)=\;\frac{b_{S}^{T}\left(\theta_{m},\phi_{m}\right)d_{l}}{c}$$(9)$$b_{iso-H}\left(\theta_{m},\phi_{m}\right)={\left[b_{H}^{T}\left(\phi_{m}\right),b_{V}^{T}\left(\theta_{m},\phi_{m}\right)\right]}^{T}$$(10)$$b_{iso-V}\left(\theta_{m},\phi_{m}\right)={\left[b_{V}^{T}\left(\theta_{m},\phi_{m}\right),{-b}_{H}^{T}\left(\phi_{m}\right)\right]}^{T}$$

When the incident signal is a fully polarized wave, and its horizontally polarized component is coherent with its vertically polarized component, then,(11)$$s_{H+V,m}\left(t\right)=\;\underset{p_{\gamma_{m},\eta_{m}}}{\underbrace{\left[\begin{array}{c} \cos\gamma_{m}\\ \sin\gamma_{m}e^{j\eta_{m}} \end{array}\right]}}s_{m}\left(t\right)$$

In Equation (11), $p_{\gamma_{m},\eta_{m}}$ represents the polarization vector of the mth incident signal; $\gamma_{m}$ and $\eta_{m}$ are respectively represent the polarization auxiliary angle and the polarization phase difference of the mth incident signal, ${0^{{}^{\circ}}\leq \gamma }_{m}\leq 90^{{}^{\circ}}$, ${{-180}^{{}^{\circ}}\leq \eta }_{m}\leq 180^{{}^{\circ}}$.

In summary, the received signal of a multi-polarized array can be expressed as(12)$$x\left(t\right)=\underset{b_{1}}{\underbrace{H_{\theta_{1},\phi_{1}}p_{\gamma_{1},\eta_{1}}}}s_{1}\left(t\right)+i\left(t\right)+n\left(t\right)$$

## 3. Interference Suppression Method

The method for suppressing main-lobe non-stationary interference based on a multi-polarized array is shown in Figure 2. The transmitted linear frequency modulated (LFM) signal, after being reflected by the ionosphere, is received by the multi-polarized array. Subsequently, a reference signal is constructed to perform matched filtering on the echo signals received by each channel. Peaks are then selected from the results of the matched filtering, and the selection results from each channel are stacked to form a column vector, which can be used as the steering vector of the target during subsequent beamforming. Meanwhile, time–frequency analysis is performed on the received signals of each channel, followed by reconstructing the time–frequency covariance matrix by selecting high-energy time–frequency points. Finally, beamforming is conducted using the target steering vector and the time–frequency covariance matrix to obtain the result after interference suppression. This section explains the processes of target steering vector determination, time–frequency covariance matrix construction, and beamforming, respectively.

### 3.1. Calculating Target Steering Vector

Assuming that the transmitted signal $s_{1}\left(t\right)$ is a known LFM signal, and given that the interference and target signals are incoherent with each other, where $\tau_{0}$ represents the delay from transmission to reception after reflection, by performing matched filtering on the received signal of each channel, we can have:(13)$$x\left(t\right)\times s_{1}^{*}\left(-t\right)=\;b_{1}s_{1}\left(t\right)\times s_{1}^{*}\left(-t\right)\;+\;i\left(t\right)\times s_{1}^{*}\left(-t\right)\;+\;n\left(t\right)\times s_{1}^{*}\left(-t\right)$$

In Equation (13), $s_{1}\left(t\right)\times s_{1}^{*}\left(-t\right)\propto sinc\left(B\left(t-\tau_{0}\right)\right)e^{j\omega_{0}\tau_{0}}$ and B represent the bandwidth of the LFM signal, and both interference and noise are assumed to be incoherent with the target signal, we can select the output of the matched filter when $t=\tau_{0}$, denoted as $\alpha_{0}$. At this point, both $i\left(t\right)\times s_{1}^{*}\left(-t\right)$ and $n\left(t\right)\times s_{1}^{*}\left(-t\right)$ are approximately equal to 0. Therefore, it can be approximated that ${\hat{b}}_{1}={\alpha_{0}b}_{1}$. The definition of $b_{1}$ is given in Equation (12).

It should be noted that this method is greatly affected by the signal-to-noise ratio (SNR). When the SNR is very low, $n\left(t\right)\times s_{1}^{*}\left(-t\right)$ cannot be approximated as 0, and there will be errors in the obtained target steering vector, which can severely impact the subsequent beamforming processes.

### 3.2. Constructing Time–Frequency Covariance Matrix

Performing a short-time Fourier transform (STFT) on Equation (12) can yield:(14)$$X\left(t,f\right)\;=\;\sum_{m=1}^{M}\underset{b_{m}}{\underbrace{H_{\theta_{m},\phi_{m}}p_{\gamma_{m},\eta_{m}}}}S_{m}\left(t,f\right)+N\left(t,f\right)=BS\left(t,f\right)\;+\;N\left(t,f\right)$$(15)$$B=\left[b_{1},b_{2},\ldots ,b_{M}\right]$$(16)$$S\left(t,f\right)={\left[S_{1}\left(t,f\right),S_{2}\left(t,f\right),\ldots ,S_{M}\left(t,f\right)\right]}^{T}$$(17)$$X\left(t,f\right)={\left[X_{1}\left(t,f\right),X_{2}\left(t,f\right),\ldots ,X_{6L}\left(t,f\right)\right]}^{T}$$(18)$$N\left(t,f\right)={\left[N_{1}\left(t,f\right),N_{2}\left(t,f\right),\ldots ,N_{6L}\left(t,f\right)\right]}^{T}$$(19)$$X_{l}\left(t,f\right)=\int_{-\infty }^{+\infty }x_{l}\left(\tau \right)g\left(\tau -t\right)e^{-j2\pi f\tau }d\tau$$(20)$$S_{m}\left(t,f\right)=\int_{-\infty }^{+\infty }s_{m}\left(\tau \right)g\left(\tau -t\right)e^{-j2\pi f\tau }d\tau$$(21)$$N_{l}\left(t,f\right)=\int_{-\infty }^{+\infty }n_{l}\left(\tau \right)g\left(\tau -t\right)e^{-j2\pi f\tau }d\tau$$(22)$$g\left(t\right)=\left\{\begin{array}{c} 0.54-0.46\cos\frac{2\pi t}{T_{win}}\;\;\;\;\;\;0\leq t\leq T_{win}\\ 0\;\;\;\;\;\;\;\;\;\;\;\;\;\;\;\;\;\;\;\;\;\;\;else\; \end{array}\right.$$

Here, it is assumed that the antenna array is a fully polarized array, thus the lengths of $X\left(t,f\right)$ and $N\left(t,f\right)$ are 6L. $X_{l}\left(t,f\right)$, $S_{m}\left(t,f\right)$ and $N_{l}\left(t,f\right)$ respectively represent the time–frequency analysis of the lth receiving channel output, the time–frequency analysis of the mth incident signal and the time–frequency analysis of the additive noise of the lth receiving channel output, all of which are two-dimensional functions of time and frequency. The function $g\left(t\right)$ serves as a window function, generally chosen to be a Hamming window, and $T_{win}$ represents the window length. Since computers can only process discrete signals, it is necessary to use a discrete window function instead of Equation (22) when actually performing the short-time Fourier transform.

According to Equation (14), without considering noise, we have:(23)$$\underset{D_{XX}\left(t,f\right)}{\underbrace{X\left(t,f\right)X^{H}\left(t,f\right)}}\;=\;\underset{D_{SS}\left(t,f\right)}{\underbrace{BS\left(t,f\right)S^{H}\left(t,f\right)\;B^{H}}}$$

In Equation (23), the superscript H denotes the conjugate transpose. $D_{XX}\left(t,f\right)$ and $D_{SS}\left(t,f\right)$ are referred to as the spatial time–frequency distribution matrix and signal time–frequency distribution matrix, respectively. The calculation method for $D_{XX}\left(t,f\right)$ is shown below, and $D_{SS}\left(t,f\right)$ can be calculated in a similar way.(24)$$D_{XX}\left(t,f\right)=\left[\begin{array}{ccc} Y_{1,1}\left(t,f\right) & \ldots & Y_{1,6L}\left(t,f\right)\\ \vdots & \ddots & \vdots \\ Y_{6L,1}\left(t,f\right) & \ldots & Y_{6L,6L}\left(t,f\right) \end{array}\right]$$(25)$$Y_{m,n}\left(t,f\right)=X_{m}\left(t,f\right)X_{n}^{*}\left(t,f\right)$$

In practical processing, each output of the array can only obtain a limited number of snapshots. For the lth output, performing calculations according to Equation (19) will get a two-dimensional matrix ${\hat{X}}_{l}\left(t,f\right)$. Therefore, during practical processing, a cell array $D_{XX}\left(t,f\right)$ can be created where each cell contains a two-dimensional matrix. $D_{XX}\left(t,f\right)$ is calculated according to the following formula.(26)$$D_{XX}\left(t,f\right)=\left[\begin{array}{ccc} {\hat{Y}}_{1,1}\left(t,f\right) & \ldots & {\hat{Y}}_{1,6L}\left(t,f\right)\\ \vdots & \ddots & \vdots \\ {\hat{Y}}_{6L,1}\left(t,f\right) & \ldots & {\hat{Y}}_{6L,6L}\left(t,f\right) \end{array}\right]$$(27)$${\hat{Y}}_{m,n}\left(t,f\right)={\hat{X}}_{m}\left(t,f\right)\;⨀\;{\hat{X}}_{n}^{*}\left(t,f\right)$$

In Equation (27), ⨀ represents the Hadamard product. Assuming that the total length of the signal undergoing short-time Fourier transform is $N_{x}$, the number of points in the discrete window function is $N_{win}$, and the number of overlapping points between windows is $N_{over}$, then the dimension of ${\hat{X}}_{l}\left(t,f\right)$ is ${\left\lfloor T\right\rfloor \times N}_{win}$, $T=\left(N_{x}-N_{win}\right)/\left(N_{win}-N_{over}\right)+1$, where $\left\lfloor \cdot \right\rfloor$ denotes the floor function (rounding down to the nearest integer).

Before we estimate the spatial time–frequency covariance matrix ${\bar{D}}_{XX}$, the calculation of the frequency point judgment matrix $D_{judge}$ is needed:(28)$$D_{judge}=\frac{1}{6L}\sum_{l=1}^{6L}{\hat{Y}}_{l,l}\left(t,f\right)$$

$D_{judge}$ is still a two-dimensional matrix related to time and frequency, with a dimension of ${\left\lfloor T\right\rfloor \times N}_{win}$. In $D_{judge}$, for each time point $t_{k}$, we can select $f_{k}$ frequency points with high energy. The set consisting of the positions of these $f_{k}$ frequency points is denoted as $\Omega_{k}=\left\{\left(t_{k},f^{n}\right)|n=1,2,3,\ldots ,f_{k}\right\}$, and the total number of selected frequency points is $F=\sum_{k=1}^{\left\lfloor T\right\rfloor }f_{k}$. Let $\Omega =\left\{\Omega_{k}|k=1,2,3,\ldots ,\left\lfloor T\right\rfloor \right\}$ denote the set composed of these frequency points. Finally, the spatial time–frequency covariance matrix ${\bar{D}}_{XX}$ is obtained through the following formula.(29)$${\bar{D}}_{XX}=\frac{1}{F}\sum_{k=1}^{\left\lfloor T\right\rfloor }\sum_{n=1}^{f_{k}}D_{XX}\left(t_{k},f^{n}\right)$$

Using a case of fully polarized interference as an example, explain the process of obtaining ${\bar{D}}_{XX}$ and the form of ${\bar{D}}_{XX}$. According to Equation (23),(30)$$\begin{array}{ll} D_{XX}\left(t,f\right) & =X\left(t,f\right)X^{H}\left(t,f\right)\\ & =b_{1}S_{1}\left(t,f\right)S_{1}^{*}\left(t,f\right){b\;}_{1}^{H}\\ & +b_{2}S_{2}\left(t,f\right)S_{1}^{*}\left(t,f\right){b\;}_{1}^{H}\\ & +b_{1}S_{1}\left(t,f\right)S_{2}^{*}\left(t,f\right){b\;}_{2}^{H}\\ & +b_{2}S_{2}\left(t,f\right)S_{2}^{*}\left(t,f\right){b\;}_{2}^{H} \end{array}$$

For the $D_{judge}$ shown in Figure 3, frequency points need to be selected along two time–frequency lines, which belong to $S_{1}\left(t,f\right)$ and $S_{2}\left(t,f\right)$ respectively. When the selected frequency points belong to $S_{1}\left(t,f\right)$, there are $S_{2}\left(t_{k},f^{n}\right)\approx 0$ and $S_{2}^{*}\left(t_{k},f^{n}\right)\approx 0$; when the selected frequency points belong to $S_{2}\left(t,f\right)$, there are $S_{1}\left(t_{k},f^{n}\right)\approx 0$ and $S_{1}^{*}\left(t_{k},f^{n}\right)\approx 0$. In this case, we have:(31)$${\bar{D}}_{XX}=\frac{1}{F}\sum_{k=1}^{\left\lfloor T\right\rfloor }\sum_{n=1}^{f_{k}}D_{XX}\left(t_{k},f^{n}\right)={\bar{\sigma }}_{1,1}^{2}b_{1}{b\;}_{1}^{H}+{\bar{\sigma }}_{2,2}^{2}b_{2}{b\;}_{2}^{H}$$(32)$${\bar{\sigma }}_{m,n}^{2}=\frac{1}{F}\sum_{k=1}^{\left\lfloor T\right\rfloor }\sum_{n=1}^{f_{k}}S_{m}\left(t_{k},f^{n}\right)S_{n}^{*}\left(t_{k},f^{n}\right)$$

When the target and interference are time–frequency overlapped, Equation (31) will become:(33)$$\begin{array}{ll} {\bar{D}}_{XX} & =\frac{1}{F}\sum_{k=1}^{\left\lfloor T\right\rfloor }\sum_{n=1}^{f_{k}}D_{XX}\left(t_{k},f^{n}\right)\\ & ={\bar{\sigma }}_{1,1}^{2}b_{1}b_{1}^{H}+{\bar{\sigma }}_{1,2}^{2}b_{1}b_{2}^{H}\\ & +{\bar{\sigma }}_{2,1}^{2}b_{2}b_{1}^{H}+{\bar{\sigma }}_{2,2}^{2}b_{2}b_{2}^{H} \end{array}$$

At this point, it is necessary to perform a smoothing operation on ${\bar{D}}_{XX}$ to restore its full-rank characteristic before proceeding with subsequent beamforming. Here, we employ the spatial smoothing technique. Suppose that N electromagnetic vector sensors are grouped into a subarray. The following operations need to be performed:(34)$${\tilde{D}}_{XX}\;=\;\sum_{k=0}^{K\;-\;1}{\bar{D}}_{XX}(6k\;+\;1:6k\;+\;6N,6k\;+\;1:6k\;+\;6N)$$
where K = L − N + 1. Subsequently, beamforming can be performed by using the output signals from the first 6N channels, thus enabling the application of our proposed method in the case of time–frequency overlapped.

### 3.3. Beamforming

Assuming that the beamforming weight vector is w, then the signal power after beamforming is $w^{H}{\bar{D}}_{XX}w$. At this point, w can be designed based on the following criteria.(35)$$\underset{w}{\min}w^{H}{\bar{D}}_{XX}w\;\;\;s.t.\;\;w^{H}{\hat{b}}_{1}=1$$

According to Equation (31), minimizing $w^{H}{\bar{D}}_{XX}w$ is equivalent to minimizing ${\left\|w^{H}b_{2}\right\|}_{2}^{2}$, thereby automatically suppressing interference. This problem can be solved using the Lagrange multiplier method, and finally, the solution can be obtained as follows:(36)$$w={\bar{D}}_{XX}^{-1}{\hat{b}}_{1}{\left({\hat{b}}_{1}^{H}{\bar{D}}_{XX}^{-1}{\hat{b}}_{1}\right)}^{-1}$$

Since this weight value constrains $w^{H}{\hat{b}}_{1}=1$, which is related to $w^{H}b_{1}=\alpha_{0}^{-1}$, the power of the output signal after beamforming generally decreases. Therefore, it is still necessary to amplify the signal after beamforming. The amplification coefficient can be determined by comparing the time–frequency distribution of the signal before and after beamforming. In practice, not all received pulses are subject to interference. We can utilize the pulses without interference to conduct time–frequency analysis according to Equation (19) and determine the positions of the high-energy frequency points. Assuming that the set of positions formed by these high-energy frequency points is Ζ, we first calculate the coefficient $\nu_{1}$ by:(37)$$\nu_{1}=\frac{1}{6L}\sum_{l=1}^{6L}\sum_{\left(t,f\right)\epsilon Ζ}X_{l}\left(t,f\right)$$

Then, when we conduct time–frequency analysis and calculate the coefficient for the beamforming output signal $y\left(t\right)\;=\;w^{H}x\left(t\right)$, we obtain:(38)$$\nu_{2}=\sum_{\left(t,f\right)\epsilon Ζ}Y\left(t,f\right)$$

Ultimately, the amplification coefficient can be determined by $\nu_{1}/\nu_{2}$.

The overall steps of main-lobe interference suppression are as shown in Table 1.

## 4. Simulation Analysis

The suppression effect of a multi-polarized array is evaluated by receiving amplitude directional diagram and polarization matching directional diagram. The definitions of these two indicators are as follows:(39)$$B_{w}\left(\theta ,\phi \right)={\left\|{\left[w^{H}a_{H}\left(\theta ,\phi \right),w^{H}a_{V}\left(\theta ,\phi \right)\right]}^{T}\right\|}_{2}$$(40)$${PMP}_{w}\left(\theta ,\phi \right)=\frac{\left[w^{H}a_{H}\left(\theta ,\phi \right),w^{H}a_{V}\left(\theta ,\phi \right)\right]p_{\gamma_{m},\eta_{m}}}{{\left\|{\left[w^{H}a_{H}\left(\theta ,\phi \right),w^{H}a_{V}\left(\theta ,\phi \right)\right]}^{T}\right\|}_{2}}$$

The polarization matching directional diagram is related to the polarization parameters of the mth signal, so it differs for different signals. The receiving amplitude directional diagram reflects the amplitude gain of the multi-polarized array, while the polarization matching directional diagram reflects its polarization gain. The subsequent simulation uses an 8-element linear array, with specific simulation parameters shown in Table 2 below.

Under the given simulation conditions, the array factor of the single-polarized array is shown in Figure 4. From the figure, it can be observed that its Rayleigh limit (the angular distance from the main lobe’s peak to its first null) is approximately 19.45°. When the difference between the target angle and the interference angle is less than the Rayleigh limit, a super-resolution direction of arrival (DOA) estimation algorithm is required to separate them. Subsequent simulations will only consider interferences within the main lobe. Therefore, they will be conducted based on two scenarios: where the interference is located within the main lobe and where both the interference and the target are at the same angle.

To demonstrate the effectiveness of the proposed method, we compared it with the MVDR and WCPO methods. The time–frequency distributions of the target signal and interference signal during the simulation are shown in Figure 5a,b respectively.

### 4.1. The Interference Is Located Within the Main-Lobe

The target angle is (0°, 90°), and the interference angle is (−3°, 90°). The received amplitude directional diagram, polarization matching directional diagrams of the target and interference, and comparisons of time–frequency analysis before and after beamforming are shown below respectively.

From Figure 6, Figure 7, Figure 8 and Figure 9, it can be seen that the main-lobe appears around 0° in the receiving amplitude directional diagram with a gain of approximately 6 dB in Figure 6b, while the polarization matching directional diagram of the target has a gain of about −6 dB around 0° in Figure 7b, indicating that the array’s gain towards the target angle is approximately 0 dB at this time, thus preserving the target signal. The polarization matching directional diagram of the interference exhibits a null at −3° in Figure 8b, proving that the multi-polarized array suppresses interference through polarization at this point. The target signal cannot be observed in the time–frequency distribution before beamforming, but it becomes visible after beamforming, demonstrating that the proposed method can effectively suppress interference. Compared with the MVDR and WCPO method, When the target angle is assumed to be known, the polarization parameters of the target can be estimated to calculate the target’s steering vector, thereby suppressing interference. However, when the target angle is unknown, the target’s steering vector cannot be calculated, and thus interference cannot be suppressed. The proposed method can suppress interference without the estimation of the target parameters.

### 4.2. Interference Coincides with the Target

The target and interference angles are both (10°, 90°). The receiving amplitude directional diagram, polarization matching directional diagrams of the target and interference, and comparisons of time–frequency analysis before and after beamforming are shown below respectively.

From Figure 10, Figure 11, Figure 12 and Figure 13, it can be seen that the main-lobe appears at 10° in the receiving amplitude directional diagram, but the gain is approximately −2.5 dB. The gain of the target’s polarization matching pattern at 10° is about −6 dB. The reasons for this situation have been explained earlier, so signal amplification will be necessary when comparing time–frequency distributions subsequently. The interference’s polarization matching directional diagram exhibits a null at 10°, indicating that the multi-polarized array is suppressing interference through polarization at this time. The target signal cannot be observed in the time–frequency distribution before beamforming, while it becomes visible after beamforming, which also proves that the proposed method can effectively suppress interference at this point. In this case, since the target’s steering vector cannot be obtained, both the MVDR and WCPO methods will fail. Therefore, the proposed method is superior.

To verify the effect of interference suppression under different input signal-to-noise ratios (SNRs), the output signal-to-interference-plus-noise ratio (OSINR) can be calculated under various input SNRs, as defined below.(41)$$OSINR\left(w\right)=\frac{{\left|w^{H}b_{1}\right|}^{2}{\bar{\sigma }}_{1,1}^{2}}{\sum_{m=2}^{M}{\left|w^{H}b_{m}\right|}^{2}{\bar{\sigma }}_{m,m}^{2}{\left\|w\right\|}_{2}^{2}{\bar{\sigma }}_{noise}^{2}}$$

Figure 14 illustrates the variation of the OSINR with respect to the input SNR. As can be seen from the figure, as the input SNR increases, the OSINR improves. However, when the input SNR is low, due to the significant impact of noise on matched filtering, there exists an error between the calculated steering vector and the true steering vector. Therefore, the suppression effect at low SNRs is not ideal.

## 5. Conclusions

This paper proposes a main-lobe interference suppression method based on multi-polarized array reception for ionospheric non-stationary interference. This method uses matched filtering to calculate the target steering vector and reconstructs the time–frequency covariance matrix through time–frequency analysis and selection of time–frequency points. Compared with existing methods, the proposed method uses the waveform information of the transmitted signal and does not require estimation of the target’s polarization parameters. Simulation results show that when the waveform information is known, the target steering vector can be estimated through matched filtering, and the polarization difference between the target and the interference can be utilized to suppress the interference in the polarization domain. It should be noted that the proposed method in this paper is greatly affected by noise, and further improvements are needed in the future to make it applicable to low SNR.

## Figures and Tables

**Figure 1 sensors-25-06587-f001:**
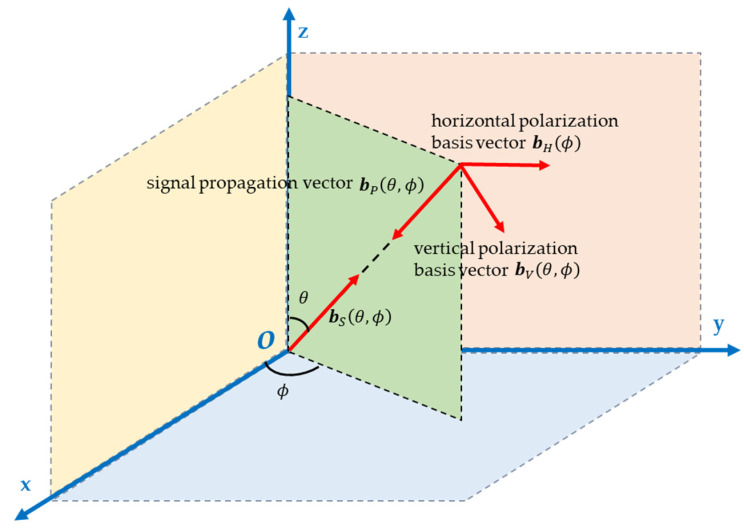
Schematic diagram of the observation coordinate system and signal wave propagation.

**Figure 2 sensors-25-06587-f002:**
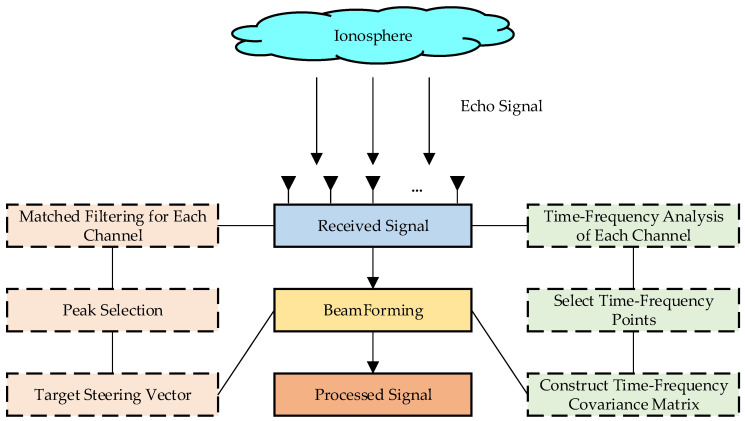
The flowchart of suppressing main-lobe non-stationary interference.

**Figure 3 sensors-25-06587-f003:**
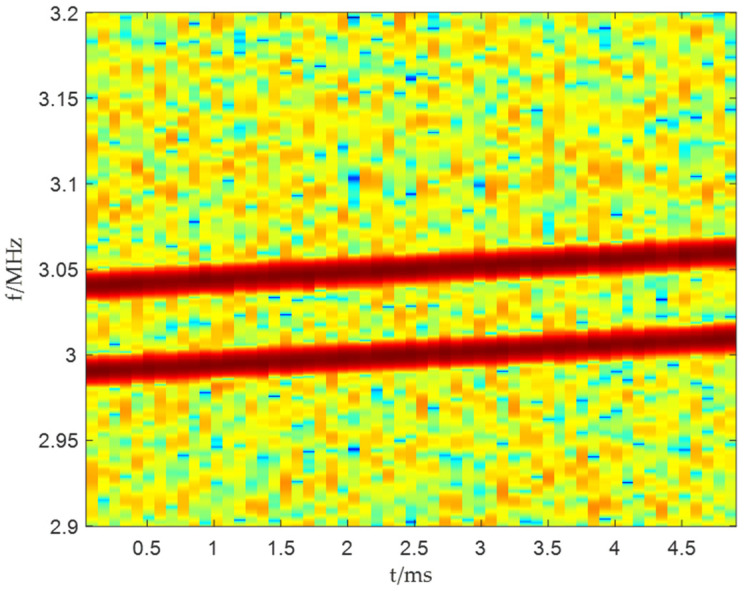
Schematic diagram of time–frequency judgment matrix.

**Figure 4 sensors-25-06587-f004:**
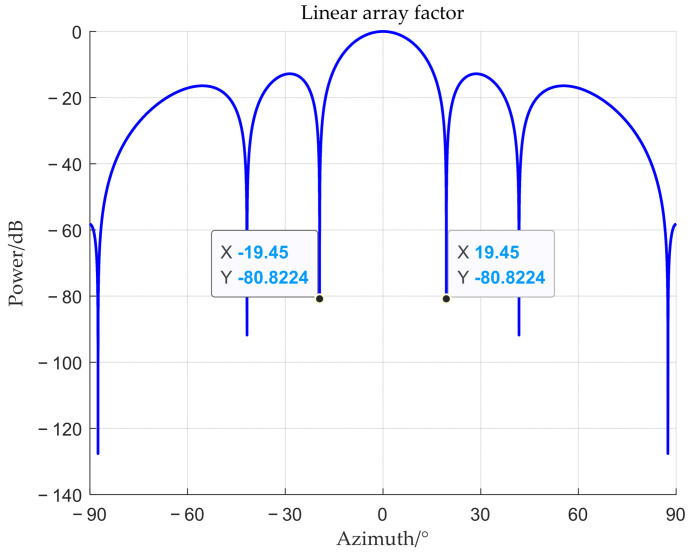
Linear array factor.

**Figure 5 sensors-25-06587-f005:**
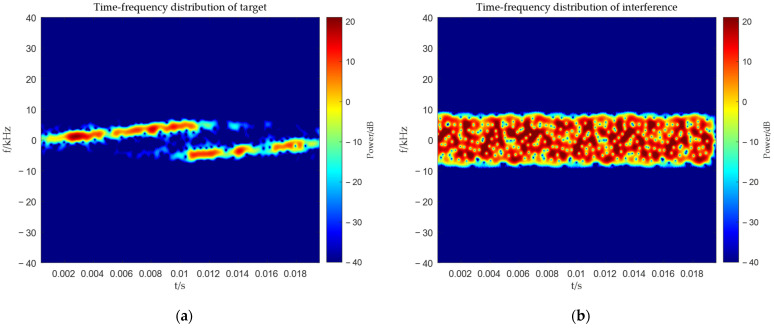
Time–frequency distribution of target and interference. (**a**) Time–frequency distribution of target; (**b**) Time–frequency distribution of interference.

**Figure 6 sensors-25-06587-f006:**
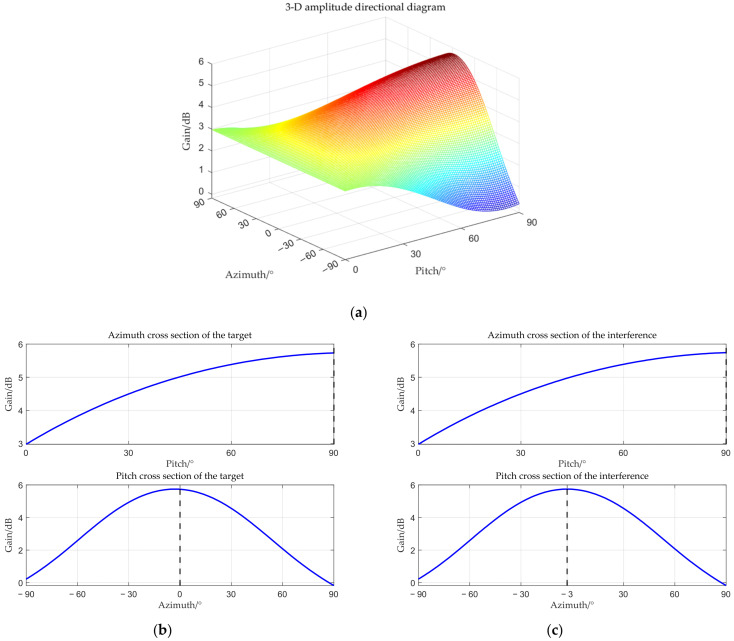
Received amplitude directional diagram. (**a**) 3-D amplitude directional diagram; (**b**) cross section of target; (**c**) cross section of interference.

**Figure 7 sensors-25-06587-f007:**
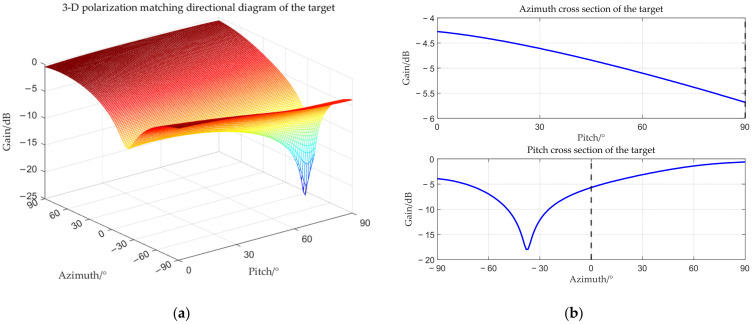
Target polarization matching directional diagram when the interference is located within the main-lobe. (**a**) 3-D polarization matching directional diagram of the target; (**b**) cross section.

**Figure 8 sensors-25-06587-f008:**
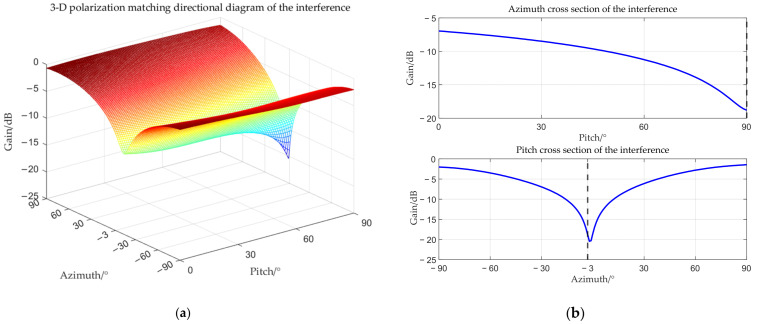
Interference polarization matching directional diagram when the interference is located within the main-lobe. (**a**) 3-D polarization matching directional diagram of the interference; (**b**) cross section.

**Figure 9 sensors-25-06587-f009:**
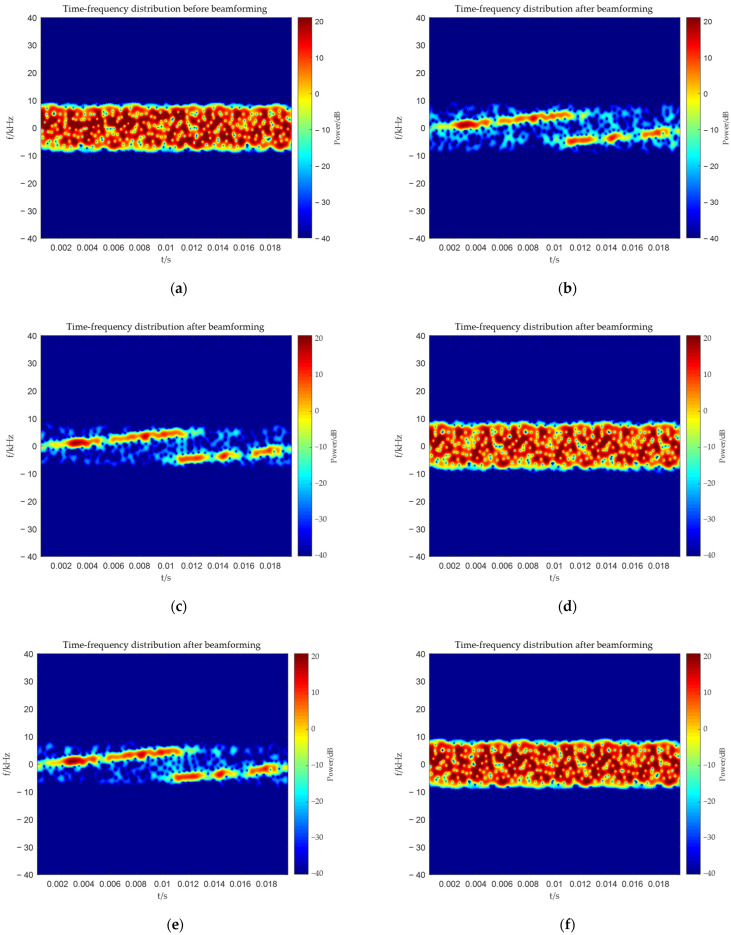
Comparison of time–frequency distribution before and after beamforming. (**a**) Before beamforming: proposed method; (**b**) after beamforming: proposed method; (**c**) after beamforming: MVDR method, incident angle of target is known; (**d**) after beamforming: MVDR method, incident angle of target is unknown; (**e**) after beamforming: WCPO method, incident angle of target is known; (**f**) after beamforming: WCPO method, incident angle of target is unknown.

**Figure 10 sensors-25-06587-f010:**
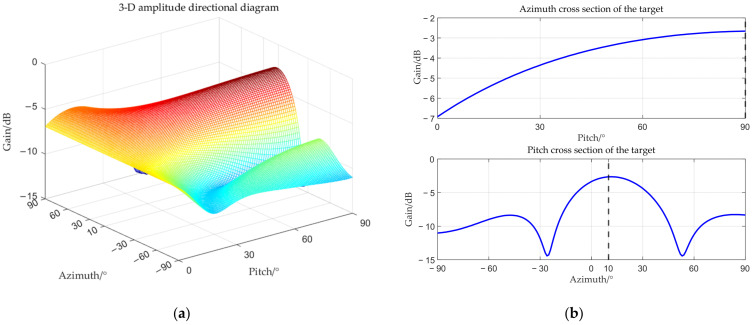
Received amplitude directional diagram. (**a**) 3-D amplitude directional diagram; (**b**) cross section of target and interference.

**Figure 11 sensors-25-06587-f011:**
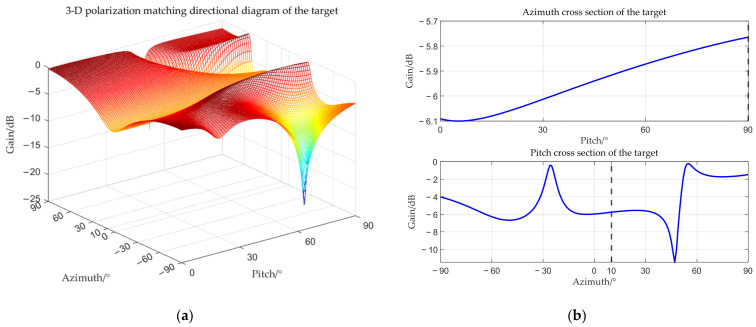
Target polarization matching directional diagram when interference coincides with the target. (**a**) 3-D polarization matching directional diagram of the target; (**b**) cross section.

**Figure 12 sensors-25-06587-f012:**
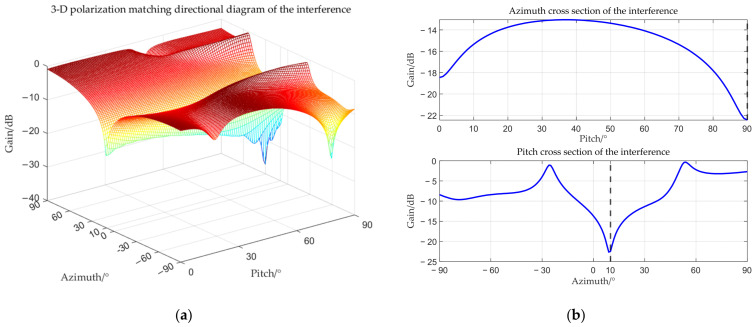
Interference polarization matching directional diagram when interference coincides with the target. (**a**) 3-D polarization matching directional diagram of the interference; (**b**) cross section.

**Figure 13 sensors-25-06587-f013:**
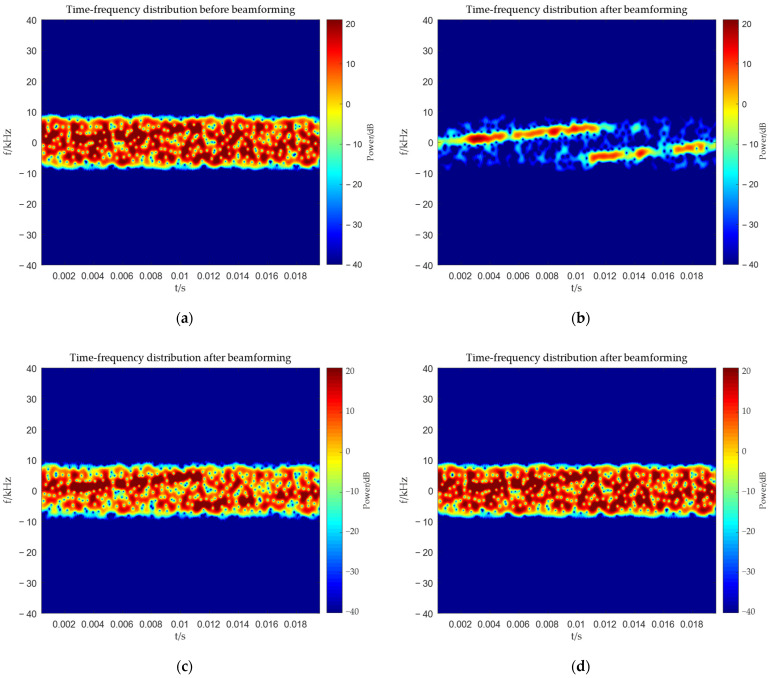
Comparison of time-frequency distribution before and after beamforming. (**a**) Before beamforming: proposed method; (**b**) after beamforming: proposed method; (**c**) after beamforming: MVDR method; (**d**) after beamforming: WCPO method.

**Figure 14 sensors-25-06587-f014:**
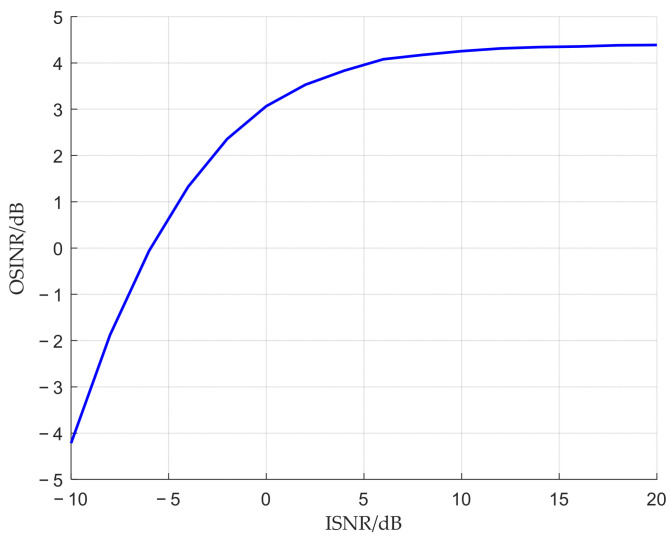
The variation of OSINR with ISNR.

**Table 1 sensors-25-06587-t001:** Main-lobe interference suppression steps.

Beamforming Method for Non-Stationary Interference in the Main-Lobe Based on a Multi-Polarized Array
1) According to Equation (13), perform matched filtering on the outputs of each channel and select the peak points at the same position to obtain the estimated target steering vector ${\hat{b}}_{1}$.
2) Based on Equations (17), (19) and (22), perform time–frequency analysis on the output of each channel of the array to obtain ${\hat{X}}_{l}\left(t,f\right)$
3) Calculating spatial time–frequency distribution matrix $D_{XX}\left(t,f\right)$ by Equations (26) and (27).
4) Determine frequency point judgment matrix $D_{judge}$ according to Equation (28), and then calculate the time–frequency covariance matrix ${\bar{D}}_{XX}$ according to Equation (29).
5) Calculating the beamforming weight vector w according to Equation (36).
6) The beamforming result is $y(t)=w^{H}x(t)$
7) Calculating amplification coefficient by Equations (37) and (38), The final result of beamforming is $y(t)=\nu_{1}w^{H}x(t)/\nu_{2}$

**Table 2 sensors-25-06587-t002:** Simulation parameters.

Parameter Name	Parameter Value
Carrier Frequency	16,320 kHz
Sampling Frequency	80 kHz
Band-Width	10 kHz
Time-Width	20 ms
Sampling Points	1600
Spacing between Array Elements	6.9 m
Number of Array Elements	8
SNR	10 dB
Fully Polarized Parameters (Target)	(20°, 10°)
Fully Polarized Parameters (Interference)	(30°, 40°)

## Data Availability

The data in this paper are all obtained through simulation, and the simulation parameters of the model are given in this paper.

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
