# Peer review of "Method for Suppressing Non-Stationary Interference in the Main-Lobe Based on a Multi-Polarized Array"

_sensors, 2025, doi:10.3390/s25216587_

Round 1
Reviewer 1 Report
Comments and Suggestions for Authors
This paper proposes a novel method for suppressing non-stationary, main-lobe interference using a multi-polarized array. The approach leverages matched filtering to obtain the target steering vector and time-frequency analysis to construct the covariance matrix. The simulation results provide a preliminary validation of the method's effectiveness under specific conditions.
However, the manuscript has several issues regarding methodological rigor and the clarity of key technical details that need to be addressed.
1. A major concern is the method for constructing the interference-plus-noise covariance matrix. The procedure selects "high-energy" time-frequency points to form the matrix, but it does not specify how to differentiate between energy originating from the target and that from the interference. Since the desired signal may also be a high-energy component, its inclusion in the covariance matrix estimation will inevitably lead to signal self-nulling, causing the beamformer to suppress the target.
2. For scenarios with time-frequency overlap between the target and interference, the paper states the necessity of a "smoothing operation" to restore the full-rank property of the covariance matrix. However, the manuscript fails to specify the exact smoothing technique used, nor does it provide a justification for choosing this approach over other methods for handling rank-deficient matrices.
3. The simulation section is limited to demonstrating the proposed method's performance in isolation. To properly assess the method's efficacy and novelty, a comparative analysis against other established main-lobe interference suppression techniques is essential.
4. The paper's definition of the "Rayleigh limit" appears to be non-standard. The author defines it as "the distance between two nulls of the beam's main-lobe," which is conventionally known as the Null-to-Null Beamwidth. The Rayleigh resolution limit typically refers to the angular distance from the main lobe's peak to its first null.
5. The manuscript defines the transpose operation twice using two different symbols, 'T' and 'H'. This is confusing. In complex signal processing, 'H' is the standard notation for the conjugate transpose (Hermitian transpose), while 'T' denotes the transpose. The notation should be corrected to adhere to standard conventions.
6. The paper suggests that an amplification coefficient is needed after beamforming and can be "determined by comparing the time-frequency distribution of the signal before and after beamforming". However, this is only a qualitative description. A specific mathematical formula or a concrete algorithm for this calculation must be provided to ensure the method's reproducibility.
Reviewer 2 Report
Comments and Suggestions for Authors
The paper develops a method to suppress non-stationary interference in the mainlobe using multi-polarised arrays. Overall, the paper reads fine, but authors are strongly suggest to enhance the paper based on below comments.
-- in the title, please consider being more specific about your proposed method
-- in Section 1, motivation of the work is unclear; related work in multi-polarised array based interference suppression is missing
-- Sveral places in Sec.3, a group of equations are provided without proper explanations, making the paper hard to follow.
-- Sec.3 is composed of a few conventional techniques. authors are suggested to highlight their novelty.
-- in simulations, how is the interference non-stationary
-- the simulation results are focused on a single set of simulation parameters, which is not convicing in validating the proposed designs. also, benchmark methods seem to be insufficient.
Reviewer 3 Report
Comments and Suggestions for Authors
This paper proposes a suppression method based on multi polarization array for non-stationary main lobe interference in ionospheric detection, which combines matched filtering, time-frequency analysis, and polarization domain processing. It has certain innovation and application value. The paper structure is complete and the simulation experiment design is reasonable. However, there are significant shortcomings in the depth of literature review, clarity of algorithm expression, complexity analysis, and adequacy of comparative experiments, which require significant modifications.
- The literature review did not closely focus on the core theme of "multi polarization array suppression of main lobe interference", especially lacking a systematic review of the application of multi polarization arrays in interference suppression. For example, P1-P2 in the article cited [16] - [24] to introduce the importance of polarization information, but did not provide a detailed overview of the specific applications of multi polarization arrays in resisting main lobe interference, such as methods based on polarization eigenvalue decomposition and polarization projection. It is suggested to supplement the representative work of multi polarization arrays in radar/communication anti-interference in recent years, and clarify that most existing methods require known interference polarization parameters or rely on high SNR assumptions, thus introducing the innovation of "no polarization parameter estimation" in this article.
- The current situation summary is relatively vague and does not clearly indicate the research gaps and challenges in the specific direction of "non-stationary main lobe interference+multi polarization array". For example, in the article, P1 mentions that "most existing methods assume that the interference characteristics are relatively stable", but does not explain why these methods cannot handle non-stationary interference, nor does it emphasize the potential of multi polarization arrays in time-frequency non-stationary scenarios. Suggest adding a paragraph at the end of the introduction to summarize the limitations of existing polarization anti-interference methods (such as feature projection and polarization filtering) in non-stationary scenarios, and point out that this article aims to use time-frequency polarization joint processing to solve this problem.
- The algorithm as a whole keeps up with the forefront, but lacks literature support for the key innovation of "matching filtering to extract polarization guidance vectors". For example, matched filtering is commonly used to extract target guidance vectors in single polarization arrays, but there is insufficient literature support for directly constructing polarization guidance vectors in multi polarization arrays. Suggest citing literature similar to using matched filtering for multi-channel signal processing (such as waveform matching techniques in MIMO radar and polarimetric radar) to enhance the theoretical basis of the method.
- The core logic of the algorithm is reasonable, but the expression is lengthy and the key steps are scattered in mathematical formulas, making it difficult for readers to grasp the main line. For example, in Section 3, matched filtering STFT、 The steps of covariance matrix construction and beamforming are described separately, lacking a comprehensive overview of the overall process. Suggest adding a sub section on algorithm overview before sections 3.1-3.3, gradually describing the algorithm process in text; Refine Figure 2 (flowchart) into more detailed sub steps, including matched filtering peak selection, time-frequency point selection criteria, weight calculation, etc.
- The flowchart is too brief and does not reflect key steps such as time-frequency point selection, covariance matrix smoothing, signal amplification, etc.
- The article completely lacks analysis of computational complexity, and does not discuss the computational burden of STFT, matrix construction, inversion, and other steps, especially the complexity increase caused by multi polarization arrays (6L channels).
- The simulation only demonstrates the effectiveness of its own method without comparing it with any mainstream algorithms, which seriously weakens the persuasiveness of the paper. For example, Figures 6-13 only show the comparison before and after the proposed method, without comparing its performance with classical methods such as polarization projection, feature projection, MVDR, etc.
Reviewer 4 Report
Comments and Suggestions for Authors
The paper introduces a novel method for suppressing non-stationary main-lobe interference in ionospheric detection. The Authors assume the use of a multi-polarized array to exploit polarization domain information without requiring estimation of target polarization parameters, thus enhancing interference suppression in realistic scenarios. The proposed approach was verified by means of computer simulations.
The paper can be improved by addressing the following issues:
- "It should be noted that the proposed method in this paper is greatly affected by noise, and further improvements are needed in the future to make it applicable to low SNR." - could you refer to any requirements regarding the method's performance? Can you point to any measurement data/research papers providing information on typical required SNR values, and assess the performance of the proposed method with respect to this requirement?
- Can you discuss the problem of computational complexity of the proposed approach? Does it limit real-time applicability of the method? Can you compare it to the alternative approaches?
- The method relies on prior knowledge of the transmitted waveform. What is the impact of waveform distortions on the method's performance?
- The use of measurement data reflecting real-world propagation conditions would strngthen the validation of the method.
- No discussion of the universality of the method has been provided. Can it be easily extended to handle different (or unknown) waveforms?
- Figure 1 - additional annotations (like incoming wave, signal propagation vector etc.) would make this sketch easier to analyze.
Reviewer 5 Report
Comments and Suggestions for Authors
Dear Authors, I am pleased to review this interesting research work in the field of non-stationary main-lobe interference mitigation.
This work proposes new interference suppression method based on multi-polarized array and proves it by simulation results. To my opinion this work seems to be interesting to the readers and well structured but some minor corrections could be made in order to increase the quality of presentation.
I recommend to add 2-3 more fresh references (dated 20242025) related to the research topic in Introduction section. It is not clear enough what other known Main-lobe interference suppression methods were proposed in the literature.
Please pay attention to the misprints (commas, spaces, etc.) throughout the manuscript (line 40, line 92, etc.).
There is no notation for “a” and “H” in equation (4).
I would recommend providing a description of all parameters used after formula (12) for the convenience of readers (Section (2)).
In Section 3 more detailed explanation of Figure 2 is needed.
Line 130: It is not clear what does “6L” mean.
There is no reference and description of Figures 3-5 in the text. Figure 6(a) - I suggest choosing a better format/location for this Figure.
During simulation Authors use antenna array with 8 elements – please give comments about this choice of simulation conditions. What simulation tool did You use?
Lines 222-230 provide a summary of the figures, but it would seem clearer to comment on each figure separately step by step.
Figure 14 illustrates the variation of OSINR with ISNR for the proposed approach, maybe it is valuable to it with known Interference suppression methods (see Line 269 in the Conclusion).
It is also recommended that the conclusion be made more precise so that it is more consistent with simulation results, given in section 4. Please prove the effectiveness of the proposed approach using clear justification based on the simulation results.
Thank You,
Sincerely, the reviewer
Round 2
Reviewer 1 Report
Comments and Suggestions for Authors
I am satisfied with the revised paper.
Author Response
Thank you for your suggestions for revision and for your affirmation of our revised article.
Reviewer 2 Report
Comments and Suggestions for Authors
while most comments are responsded well by the authors, the first comment on the title is not treated effectively
the three key words mentioned in the authors' response are more indicative about the research background/model and problem, which do not suggest your "method"
Reviewer 3 Report
Comments and Suggestions for Authors
No further comments.
Author Response

(The authors gave the same response as above.)

Reviewer 4 Report
Comments and Suggestions for Authors
Thank you for addressing the recommendations.
Author Response

(The authors gave the same response as above.)
